# Calibration Method for Relativistic Navigation System Using Parallel Q-Learning Extended Kalman Filter

**DOI:** 10.3390/s24196186

**Published:** 2024-09-24

**Authors:** Kai Xiong, Qin Zhao, Li Yuan

**Affiliations:** 1Science and Technology on Space Intelligent Control Laboratory, Beijing Institute of Control Engineering, Beijing 100094, China; 2China Academy of Space Technology, Beijing 100094, China

**Keywords:** spacecraft, autonomous navigation, relativistic navigation, Q-learning, extended Kalman filter

## Abstract

For the relativistic navigation system where the position and velocity of the spacecraft are determined through the observation of the relativistic perturbations including stellar aberration and starlight gravitational deflection, a novel parallel Q-learning extended Kalman filter (PQEKF) is presented to implement the measurement bias calibration. The relativistic perturbations are extracted from the inter-star angle measurement achieved with a group of high-accuracy star sensors on the spacecraft. Inter-star angle measurement bias caused by the misalignment of the star sensors is one of the main error sources in the relativistic navigation system. In order to suppress the unfavorable effect of measurement bias on navigation performance, the PQEKF is developed to estimate the position and velocity, together with the calibration parameters, where the Q-learning approach is adopted to fine tune the process noise covariance matrix of the filter automatically. The high performance of the presented method is illustrated via numerical simulations in the scenario of medium Earth orbit (MEO) satellite navigation. The simulation results show that, for the considered MEO satellite and the presented PQEKF algorithm, in the case that the inter-star angle measurement accuracy is about 1 mas, after calibration, the positioning accuracy of the relativistic navigation system is less than 300 m.

## 1. Introduction

Spacecraft navigation is an enabling technology for a wide variety of space missions, such as Earth satellites and deep space explorers. Currently, the commonly used navigation approach is the radio navigation based on the radio signal sent from beacons, such as ground stations and the global navigation satellite system (GNSS) [1,2]. To reduce the mission cost and improve the autonomous survival capacity, the autonomous navigation system that determines the position and velocity of the spacecraft with onboard instruments in a radio signal-denied environment is required [3,4,5]. To achieve precise navigation information for the spacecraft without the support of man-made beacons is critical for the development of future intelligent unmanned systems [6,7].

In the past few decades, several autonomous navigation techniques with different observation information sources have been studied, such as optical navigation (OPNAV) using the optical imaging of nearby celestial bodies [8,9,10], X-ray pulsar-based navigation (XNAV) [11,12,13] and star navigation based on the Doppler effect of starlight (StarNAV-DE) [14,15,16]. In the on-orbit demonstrations, the positioning accuracy of the OPNAV based on the observations of Earth is on the order of a few kilometers, while the accuracy of XNAV is less than 10 km, which is not sufficient to satisfy the high-precision navigation requirement for certain space missions. The StarNAV-DE technique has been demonstrated on the Chinese Hα Solar Explorer (CHASE). It is reported that the accuracy of the solar velocimeter observing the starlight Doppler effect on CHASE is about 2 m/s.

The spacecraft autonomous navigation method based on the relativistic perturbations of starlight is introduced in [17] and developed in [18,19]. Recently, the investigation relevant to the relativistic navigation has attracted increasing attention. A practical mathematical model to describe the relativistic perturbations to the space-based starlight observation is derived in [20]. The optical instruments for the observation of the relativistic perturbations are discussed in [21]. The application of relativistic navigation is suggested in [22] for interstellar spacecraft with high velocity such that the relativistic perturbations are not negligible. To enhance navigation accuracy and rapidity, the information fusion scheme of the relativistic navigation and the OPNAV are designed in [23,24]. The extended Kalman filter (EKF) and the unscented Kalman filter (UKF) are designed and evaluated for the implementation of the relativistic navigation in [25,26].

Among the previously mentioned autonomous navigation techniques, the relativistic navigation based on the relativistic perturbations to the inter-star angle measurement has the potential to achieve higher performance with current technology. Generally, the relativistic navigation performance depends on the measurement accuracy of the inter-star angle and the precision of the star catalog. As the inter-star angle can be measured with the accuracy of a few mas with state-of-the-art instruments, and the error of the modern star catalog is less than 0.1 mas, it is considered that relativistic navigation is a promising method to achieve high performance. In comparison with the OPNAV, an advantage of relativistic navigation is that the high-accuracy observation of starlight is generally easier than that of a nearby celestial body. Compared with the XNAV, relativistic navigation is competitive, as the number of visible stars is much more than the X-ray pulsars suitable for navigation. In addition, as the stability of the inter-star angle calculated with the star catalog is rather high, the main difficulty of the StarNAV-DE technique due to the poor stability of the stellar spectra is avoided.

In the relativistic navigation system, at least two star sensors separated from each other by a large angle are required to measure the inter-star angles, which reveal the variations in the relativistic perturbations, including stellar aberration and starlight gravitational deflection. The inter-star angle measurement bias caused by the misalignment of the star sensors strongly affects the performance of the relativistic navigation system. The main motivation of this study is to calibrate the measurement bias accurately via a fine-designed navigation filter. A common approach is to model the measurement bias as calibration parameters, which can be estimated together with the position and velocity of the spacecraft through the EKF.

It is well known that the state estimation accuracy of the EKF depends on the tuning of the process and measurement noise covariance matrices [27,28]. As the measurement noise covariance matrix can be determined through the specification of the star sensors, the problem remains in determining the process noise covariance matrix, especially for the elements related to the calibration parameters. Generally, it is difficult to obtain the optimal noise covariance matrices in the absence of exact statistical knowledge about the process noise. Several attempts have been made in the literature to develop adaptive filters [29,30,31]. For the study of autonomous navigation, the most widely used method is the adaptive extended Kalman filter (AEKF), where the noise covariance matrices are estimated together with the state vector. However, it is often difficult to guarantee the estimation accuracy of the noise covariance matrix in the presence of the state estimation error. To cope with this problem, a potential method is to combine the Q-learning approach with the EKF for tuning the process noise covariance matrix automatically [32,33,34,35,36]. The key idea of the parallel Q-learning extended Kalman filter (PQEKF) is the integration of the EKF and the Q-learning approach, where the process noise covariance matrix of the EKF is selected with the Q-learning approach, whose reward is constructed with the innovation of the EKF, such that the appropriated covariance matrix is determined to improve the filtering accuracy.

This paper studies a measurement bias calibration method for a relativistic navigation system. The main contributions of this study are as follows: (1) The PQEKF is presented to adjust the process noise covariance matrix related to the calibration parameters and that related to the position and velocity vectors, respectively. The PQEKF is different from its original version presented in [24] in that two learning agents are designed to work in parallel such that the flexibility of the algorithm is improved. (2) It is illustrated that the PQEKF is effective for calibrating the inter-star angle measurement bias of the star sensors. The simulation shows that, after calibration, the relativistic navigation accuracy for the MEO satellite is on the order of 300 m in the case that the standard deviation of the measurement noise is about 1 mas. (3) The principle of the presented method can be further applied to cope with other state estimation problems that require autonomous parameters tuning.

The remaining part of the paper is organized as follows: Section 2 formulates the mathematical model of the relativistic navigation system. Section 3 presents the PQEKF algorithm for the relativistic navigation system to estimate the calibration parameters. Section 4 evaluates the performance of the navigation filter via simulations. Finally, Section 5 concludes the paper.

## 2. Relativistic Navigation System Model

### 2.1. Basic Principle of Relativistic Navigation

The concept of the relativistic navigation is illustrated in Figure 1.

The basic principle is to estimate the position and velocity of the spacecraft based on the inter-star angle measurement, which is related to the relativistic perturbations including the stellar aberration and the starlight gravitational deflection. Considering that the effect of the stellar aberration and the starlight gravitational deflection to the inter-star angle measurement can be written as the function of the spacecraft velocity and position, respectively, the velocity and position information of the spacecraft can be extracted from the inter-star angle measurement. The derivation about the effect of relativistic perturbations to starlight is shown in [17]. As a basic of the navigation filter design, the observability analysis of the relativistic navigation system can be found in [24]. The spacecraft attitude determination method based on the stellar measurement is omitted here as it has been widely studied in literature, such as in [37].

As shown in Figure 2, a group of star sensors on a rigid platform are used to obtain the inter-star angle measurement. Then, the navigation filter is implemented to incorporate the orbital dynamics and the inter-star angle measurement, such that the position and velocity of the spacecraft are estimated. As the inter-star angle measurement bias caused by the misalignment of various star sensors may seriously degrade the navigation performance, the calibration method is required to estimate and compensate for the measurement bias. With accurate star sensors, a precise star catalog and a high-fidelity orbital dynamics model, high navigation performance is achievable if the calibration method is designed appropriately.

### 2.2. Dynamic Model

The dynamic model and the measurement model of the relativistic navigation system are constructed for the design of the navigation filter. The state vector is composed of the position vector $r_{k}={\left[\begin{array}{ccc} r_{xk} & r_{yk} & r_{zk} \end{array}\right]}^{T}$, the velocity vector $v_{k}={\left[\begin{array}{ccc} v_{xk} & v_{yk} & v_{zk} \end{array}\right]}^{T}$ and the calibration parameter vector $b_{k}={\left[\begin{array}{ccc} \cdots & b_{ijk} & \cdots \end{array}\right]}^{T}$, which is shown as
(1)$$x_{k}={\left[\begin{array}{ccc} r_{k}^{T} & v_{k}^{T} & b_{k}^{T} \end{array}\right]}^{T}$$
where the sub-label *k* denotes the discrete time, and *i* and *j* are used to distinguish different stars. The position and velocity vectors are defined in the Earth-centered inertial coordinate system.

For the Earth satellite, the dynamic model to describe the time evolution of the state vector is written as
(2)$$x_{k}=f\left(x_{k-1}\right)+w_{k}$$
with
(3)$$f\left(x_{k}\right)=x_{k}+\phi (x_{k})\tau$$
(4)$$\phi \left(x_{k}\right)=\left[\begin{array}{c} v_{k}\\ -\frac{\mu_{E}r_{k}}{{\left\|r_{k}\right\|}^{3}}+p(r_{k})\\ 0_{m\times 1} \end{array}\right]$$
where τ is the step size, $\mu_{E}$ is the gravitational constant of Earth and $\left\|\cdot \right\|$ denotes the Euclidean norm of a vector. The function $p(r_{k})$ denotes the accelerations of the spacecraft generated by perturbations other than the central gravity of Earth, such as the non-spherical gravity of Earth, atmospheric drag, sunlight pressure and lunar and solar gravity. The expression of the function $p(r_{k})$ can be found in [1]. The un-modeled perturbations are considered to be relatively small and can be absorbed in the process noise $w_{k}$, which is assumed as a zero-mean noise with the covariance matrix $Q_{k}$. $Q_{k}$ as a symmetric and positive definite matrix.

### 2.3. Measurement Model

Considering the stellar aberration, the starlight gravitational deflection and the measurement bias, when multiple stars are observed simultaneously, the measurement model to describe the relation between the inter-star angle measurement and the state vector is formulated as
(5)$$y_{k}=h\left(x_{k}\right)+\nu_{k}$$
with
(6)$$y_{k}=\left[\begin{array}{c} \vdots \\ y_{ijk}\\ \vdots \end{array}\right],\;h\left(x_{k}\right)=\left[\begin{array}{c} \vdots \\ h_{ij}(x_{k})\\ \vdots \end{array}\right]$$
where $y_{k}$ is the measurement vector, $\nu_{k}$ is the measurement noise with a covariance matrix $R_{k}$, $R_{k}$ is a symmetric and positive definite matrix and $y_{ijk}$ is the measurement of the angle between two stars distinguished by i and j. The elements $h_{ij}(x_{k})$ in the function vector $h\left(x_{k}\right)$ are expressed as
(7)$$h_{ij}\left(x_{k}\right)={\left(u_{Iik}^{\prime }\right)}^{T}u_{Ijk}^{\prime }+\frac{1}{c}\left[1-{\left(u_{Iik}^{\prime }\right)}^{T}u_{Ijk}^{\prime }\right]\left({\left(v_{k}+v_{E,k}\right)}^{T}u_{Iik}^{\prime }+{\left(v_{k}+v_{E,k}\right)}^{T}u_{Ijk}^{\prime }\right)-\;\;\frac{1}{\;c^{2}}\left[1-{\left(u_{Iik}^{\prime }\right)}^{T}u_{Ijk}^{\prime }\right][{\left({\left(v_{k}+v_{E,k}\right)}^{T}u_{Iik}^{\prime }\right)}^{2}+{\left({\left(v_{k}+v_{E,k}\right)}^{T}u_{Ijk}^{\prime }\right)}^{2}+\;\left({\left(v_{k}+v_{E,k}\right)}^{T}u_{Iik}^{\prime }\right)\left({\left(v_{k}+v_{E,k}\right)}^{T}u_{Ijk}^{\prime }\right)-\begin{array}{c} {\left(v_{k}+v_{E,k}\right)}^{T}(v_{k}+v_{E,k})] \end{array}+b_{ijk}$$
where c is the speed of light, $v_{E,k}$ is the velocity vector of Earth’s center relative to the solar system barycenter (SSB) and $u_{Iik}^{\prime }$ is the line-of-sight (LOS) vector of the ith star as seen by a fictitious stationary observer. The expression of $u_{Iik}^{\prime }$ is given by
(8)$$u_{Iik}^{\prime }=u_{Iik}+\delta u_{Iik}$$
where $u_{Iik}$ is the unit LOS vector of the ith star in the absence of the gravitational field, which is calculated from the star catalog. $\Delta u_{Iik}$ denotes the effect of Earth’s gravitational field to the unit LOS vector of the ith star, which is described as
(9)$$\delta u_{Iik}=\frac{2\mu_{E}}{c^{2}}\frac{\left(1-u_{Iik}^{T}r_{k}/\left\|r_{k}\right\|\right)(I_{3\times 3}-u_{Iik}u_{Iik}^{T})r_{k}}{{\left\|(I_{3\times 3}-u_{Iik}u_{Iik}^{T})r_{k}\right\|}^{2}}$$

Note that the measurement bias $b_{ijk}$ is modeled as the calibration parameter. The derivation of the Jacobian matrix $H_{k}$ for the measurement model shown in (5) is similar to that in [24].

## 3. Navigation Filtering Algorithm

### 3.1. Extended Kalman Filter

The extended Kalman filter (EKF) is one of the most applied navigation filtering algorithms. The traditional EKF can be adopted as the navigation filter to estimate the state vector $x_{k}$ based on the measurement vector $y_{k}$. For clarity, the equations of the EKF are summarized in Algorithm 1.
**Algorithm 1**: Extended Kalman filter.$1:\;\mathrm{function}\;EKF({\hat{x}}_{k-1},P_{k-1},y_{k},Q_{k},R_{k})$$2:\;{\hat{x}}_{k|k-1}\leftarrow f({\hat{x}}_{k-1})$     ⊳ prediction$3:\;P_{k|k-1}\leftarrow F_{k}P_{k-1}F_{k}^{T}+Q_{k}$$4:\;K_{k}\leftarrow P_{k|k-1}H_{k}^{T}{\left(H_{k}P_{k|k-1}H_{k}^{T}+R_{k}\right)}^{-1}$$5:\;{\tilde{y}}_{k}\leftarrow y_{k}-h({\hat{x}}_{k|k-1})$$6:\;{\hat{x}}_{k}\leftarrow {\hat{x}}_{k|k-1}+K_{k}{\tilde{y}}_{k}$     ⊳ update$7:\;P_{k}\leftarrow \left(I-K_{k}H_{k}\right)P_{k|k-1}{\left(I-K_{k}H_{k}\right)}^{T}+K_{k}R_{k}K_{k}^{T}$$8:\;\mathrm{return}\;{\hat{x}}_{k}$$,\;P_{k}$$,\;{\tilde{y}}_{k}$9: **end function**

In the algorithm, ${\hat{x}}_{k|k-1}$ and ${\hat{x}}_{k}$ are the predicate and the estimate of the state vector, $P_{k|k-1}$ and $P_{k}$ are their corresponding estimation error covariance matrices, $K_{k}$ is the Kalman gain, ${\tilde{y}}_{k}$ is the measurement innovation and $F_{k}={\left.\frac{\partial f}{\partial x}\right|}_{x={\hat{x}}_{k-1}}$ and $H_{k}={\left.\frac{\partial h}{\partial x}\right|}_{x={\hat{x}}_{k|k-1}}$ are the Jacobian matrices.

In the EKF algorithm, the state prediction ${\hat{x}}_{k|k-1}$ is updated with the measurement $y_{k}$, where the strength of the state update is controlled with the Kalman gain $K_{k}$. In essence, the estimation accuracy of the EKF depends on the system model and filtering parameters. It is seen from Algorithm 1 that the tuning of the filtering parameters, such as the noise covariance matrices $Q_{k}$ and $R_{k}$, plays an important role in optimizing the Kalman gain $K_{k}$, such that the observation information in $y_{k}$ is extracted adequately, while the measurement noise $\nu_{k}$ is suppressed effectively. In practice, it is often difficult to design the optimal noise covariance matrices in the absence of exact statistical knowledge about $w_{k}$ and $\nu_{k}$. In order to guarantee navigation performance, it is a worthy research field to study how to set and adjust the noise covariance matrices appropriately.

According to practical experience, it is often inefficient to tune the process noise covariance matrix together with the measurement noise covariance matrix. For the considered navigation system, the measurement noise covariance matrix can be determined according to the measurement accuracy specification of the star sensors. However, less experience is inheritable for the aerospace engineers to fine tune the process noise covariance matrix related to the calibration parameters. Thus, the adaptation of the process noise covariance matrix $Q_{k}$ is studied in this paper.

### 3.2. Q-Learning Approach

In recent years, reinforcement learning (RL) has received considerable attention, with many successful applications in various fields, such as computer science, robotics systems and control engineering [38,39]. RL is becoming a major tool in the field of artificial intelligence, such that an agent can make their own choice in an operational environment without an environmental model or labeled data. Q-learning is a representative RL approach and many studies have described its uses in solving different problems [40,41,42,43]. The combination of the EKF and Q-learning is a promising direction as both are familiar to aerospace engineers and easy to implement on the spacecraft with limited computational power.

This paper presents a parallel Q-learning extended Kalman filter (PQEKF), where the Q-learning approach is introduced to select the appropriate process noise covariance matrix through its trial-and-error mechanism, which helps to improve the filtering performance. This method differs from the Q-learning extended Kalman filter (QLEKF) presented in [24] in that two parallel learning agents, owning their separated state space, are designed for adjusting the process noise covariance corresponding to the kinematic state and the calibration parameters, respectively, which improves the flexibility of the algorithm.

In Q-learning, the agent interacts with the environment iteratively to learn the optimal strategy. The learning agent’s strategy is contained in a Q-table, which is composed of the Q-function $Q^{\left(i\right)}(s,a)$ for the state s∈S and the action a∈A, where S and A are the state space and the action space, $i\in Z_{+}$ denotes the number of iterations and $Z_{+}$ represents the set of positive integer numbers. In each iteration, the agent performs an action a in the state s according to the current strategy and receives feedback, such as a utility function U(s,a), from the environment, which indicates whether the strategy is good or not. Then, the Q-function $Q^{\left(i\right)}(s,a)$ of the agent in the Q-table is updated based on the utility function. The strategy contained in the Q-table will be optimized when sufficient iterations are implemented.

For convenience, the iterative update equation of the Q-function is shown as follows
(10)$$Q^{\left(i\right)}\left(s,a\right)=U\left(s,a\right)+\gamma Q^{\left(i-1\right)}\left(s^{\prime },a^{\left(i-1\right)}(s^{\prime })\right)$$
with
(11)$$a^{\left(i-1\right)}\left(s^{\prime }\right)=arg\;\underset{a^{\prime }\in A}{\min}Q^{\left(i-1\right)}\left(s^{\prime },a^{\prime }\right)$$
where $Q^{\left(i\right)}\left(s,a\right)$ is the Q-function that the agent gained at the *i*-th iteration, $U\left(s,a\right)$ is the current utility function obtained from the environment, γ∈[0, 1) is the discounted factor, which is introduced for the tradeoff between the current utility function and the cumulated utility function, and $s^{\prime }$ is the state after action a is performed.

The convergence of the Q-learning approach and the stability of the QLEKF algorithm are analyzed in [32,34], respectively. In this paper, a concise error-bound analysis of the Q-function $Q^{\left(i\right)}\left(s,a\right)$ considering the finite number of iterations and calculation error is summarized in Theorems 1–3, which are helpful for the readers to grasp the key idea of the iterative update process shown in (10).


**Theorem** **1.***Considering the iterative update equation shown in (10), if the initial Q-function* $Q^{\left(0\right)}\left(s,a\right)$*satisfies the following condition for all s∈S and a∈A,*(12)$$Q^{\left(0\right)}\left(s,a\right)\geq U\left(s,a\right)+\gamma Q^{\left(0\right)}\left(s^{\prime },a^{\left(0\right)}(s^{\prime })\right)$$
and
(13)$$Q^{\left(i\right)}\left(s,a\right)\geq 0,$$
then
(14)$$\underset{i\to \infty }{\lim}Q^{\left(i\right)}\left(s,a\right)=Q^{*}\left(s,a\right)$$
where
(15)$$Q^{*}\left(s,a\right)=U\left(s,a\right)+\gamma Q^{*}\left(s^{\prime },a^{*}(s^{\prime })\right)$$
with
(16)$$a^{*}\left(s^{\prime }\right)=arg\;\underset{a^{\prime }\in A}{\min}Q^{*}\left(s^{\prime },a^{\prime }\right)$$


The proof of the theorem is collected in Appendix A. It indicates that, under certain conditions, the current Q-function $Q^{\left(i\right)}\left(s,a\right)$ is convergent to the optimal Q-function $Q^{*}\left(s,a\right)$ with an infinite number of iterations. In fact, the iterative update equations shown in (10) and (11) can be combined as
(17)$$Q^{\left(i\right)}\left(s,a\right)=U\left(s,a\right)+\gamma \underset{a^{\prime }\in A}{\min}Q^{\left(i-1\right)}\left(s^{\prime },a^{\prime }\right).$$
Correspondingly, the optimal Q-function is rewritten as
(18)$$Q^{*}\left(s,a\right)=U\left(s,a\right)+\gamma \underset{a^{\prime }\in A}{\min}Q^{*}\left(s^{\prime },a^{\prime }\right).$$

The following lemmas are useful to analyze the error bounds of the current Q-function in the presence of calculation error with a finite number of iterations.


**Lemma** **1.***Considering the current Q-function sequence* $\left\{Q_{1}^{\left(i\right)}\left(s,a\right)\right\}$ *and $\left\{Q_{2}^{\left(i\right)}\left(s,a\right)\right\}$ obtained from (10), if the following inequality holds for all s∈S and a∈A*,


(19)$$Q_{1}^{\left(0\right)}\left(s,a\right)\leq Q_{2}^{\left(0\right)}\left(s,a\right),$$
then for $i\in Z_{+}$, the following inequality is satisfied
(20)$$Q_{1}^{\left(i\right)}\left(s,a\right)\leq Q_{2}^{\left(i\right)}\left(s,a\right).$$


**Lemma** **2.***Considering the current Q-function* $Q^{\left(i\right)}\left(s,a\right)$*shown in (17) and a scalar ϵ, let*


(21)$${\bar{Q}}^{\left(i\right)}\left(s,a\right)=Q^{\left(i\right)}\left(s,a\right)+\epsilon$$
and define the Q-learning operator $L^{\left(1\right)}$
(22)$$(L^{\left(1\right)}{\bar{Q}}^{\left(i\right)})\left(s,a\right)=U\left(s,a\right)+\gamma \;\underset{a^{\prime }\in A}{\min}{\bar{Q}}^{\left(i\right)}\left(s^{\prime },a^{\prime }\right).$$
The composition of the mapping with itself t times is denoted by $L^{\left(t\right)}$. Then for $i\in Z_{+}$, the following equality is satisfied
(23)$$(L^{\left(t\right)}{\bar{Q}}^{\left(i\right)})\left(s,a\right)=Q^{\left(i+t\right)}\left(s,a\right)+\gamma^{t}\epsilon .$$

The proofs of Lemmas 1 and 2 are collected in Appendix B and Appendix C. With these prerequisites, the error bound of $Q^{\left(i\right)}\left(s,a\right)$ with a finite number of iterations is summarized in the following theorem.


**Theorem** **2.***For the current Q-function* $Q^{\left(i\right)}\left(s,a\right)$*shown in (17) and the optimal Q-function $Q^{*}\left(s,a\right)$ shown in (18), if the conditions shown in Theorem 1 hold, then the following inequality is satisfied for all s∈S, a∈A and $i\in Z_{+}$*


(24)$$\frac{\gamma }{1-\gamma }{\underline{\epsilon }}_{i}\leq Q^{*}\left(s,a\right)-Q^{\left(i\right)}\left(s,a\right)\leq \frac{\gamma }{1-\gamma }{\bar{\epsilon }}_{i}$$
where
(25)$${\underline{\epsilon }}_{i}=\underset{s\in S,a\in A}{\min}\left[Q^{\left(i\right)}\left(s,a\right)-Q^{\left(i-1\right)}\left(s,a\right)\right]$$
(26)$${\bar{\epsilon }}_{i}=\underset{s\in S,a\in A}{\max}\left[Q^{\left(i\right)}\left(s,a\right)-Q^{\left(i-1\right)}\left(s,a\right)\right].$$

The proof of Theorem 2 is collected in Appendix D. It is seen from Theorem 2 that the bound of the error between the current Q-function and the optimal Q-function is determined by the difference between $Q^{\left(i\right)}\left(s,a\right)$ and $Q^{\left(i-1\right)}\left(s,a\right)$ with a finite number of iterations. According to Theorem 1, the error bound defined in (25) and (26) tends to zero as i→∞.

In addition, consider a calculated Q-function ${\hat{Q}}^{\left(i\right)}\left(s,a\right)$ with calculation error. Based on Theorem 2, the bound of the error between the calculated Q-function ${\hat{Q}}^{\left(i\right)}\left(s,a\right)$ and the optimal Q-function $Q^{*}\left(s,a\right)$ with a finite number of iterations is studied in the following theorem.


**Theorem** **3.***Considering the calculated Q-function sequence* $\left\{{\hat{Q}}^{\left(i\right)}\left(s,a\right)\right\}$*, if the following inequality holds for all*s∈S*,*a∈A *and* $i\in Z_{+}$


(27)$$\left|{\hat{Q}}^{\left(i\right)}\left(s,a\right)-Q^{\left(i\right)}\left(s,a\right)\right|\leq \varepsilon_{i}(s,a)$$
where $\varepsilon_{i}(s,a)$ is the function that describes the error bound between the calculated Q-function and the current Q-function, and the conditions shown in Theorem 1 hold, then the following inequality is satisfied
(28)$$-{\underline{c}}_{i}\leq Q^{*}\left(s,a\right)-{\hat{Q}}^{\left(i\right)}\left(s,a\right)\leq {\bar{c}}_{i}$$
with
(29)$${\underline{c}}_{i}=\frac{{\bar{\varepsilon }}_{i}+\gamma {\bar{\varepsilon }}_{i-1}}{1-\gamma }-\frac{\gamma }{1-\gamma }\underset{s\in S,a\in A}{\min}\left[{\hat{Q}}^{\left(i\right)}\left(s,a\right)-{\hat{Q}}^{\left(i-1\right)}\left(s,a\right)\right]$$
(30)$${\bar{c}}_{i}=\frac{{\bar{\varepsilon }}_{i}+\gamma {\bar{\varepsilon }}_{i-1}}{1-\gamma }+\frac{\gamma }{1-\gamma }\underset{s\in S,a\in A}{\max}\left[{\hat{Q}}^{\left(i\right)}\left(s,a\right)-{\hat{Q}}^{\left(i-1\right)}\left(s,a\right)\right]$$
where
(31)$${\bar{\varepsilon }}_{i}=\underset{s\in S,a\in A}{max}\varepsilon_{i}(s,a).$$

The proof of Theorem 3 is collected in Appendix E. It is seen from Theorem 3 that the error bound of the calculated Q-function ${\hat{Q}}^{\left(i\right)}\left(s,a\right)$ is determined by the calculated error ${\bar{\varepsilon }}_{i}$ and the difference between ${\hat{Q}}^{\left(i\right)}\left(s,a\right)$ and ${\hat{Q}}^{\left(i-1\right)}\left(s,a\right)$ with a finite number of iterations. It is seen from the theorem that the error bound defined in (29) and (30) tends to zero if the calculation error is zero and i→∞.

### 3.3. Parallel Q-Learning Extended Kalman Filter

In this sub-section, the PQEKF algorithm is presented based on the EKF shown in Algorithm 1 and the iterative update equation of Q-learning shown in (10). To fine tune the process noise covariance matrix of the filter, it is assumed that $Q_{k}$ is a diagonal matrix with the following structure
(32)$$Q_{k}=\left[\begin{array}{cc} Q_{rv,k} & 0\\ 0 & Q_{b,k} \end{array}\right]$$
where $Q_{rv,k}$ is the sub-matrix corresponding to the position $r_{k}$ and velocity $v_{k}$ in the state vector, while $Q_{b,k}$ is the sub-matrix corresponding to the calibration parameter vector $b_{k}$. To avoid the curse of dimensionality, two parallel learning agents are designed to tune the sub-matrices $Q_{rv,k}$ and $Q_{b,k}$ individually in an episode. Therefore, two Q-tables, each for one learning agent, should be updated in the algorithm. The Q-functions of the two agents are expressed as $Q_{rv}\left(s_{rv},a_{rv}\right)$ ($s_{rv}\in S_{rv}$, $a_{rv}\in A_{rv}$) and $Q_{b}\left(s_{b},a_{b}\right)$ ($s_{b}\in S_{b}$, $a_{b}\in A_{b}$), respectively, where $S_{rv}$, $S_{b}$ and $A_{rv}$, $A_{b}$ are the corresponding state space and action space. Note that the superscript $\left(i\right)$ of the Q-function is omitted here for simplicity.

Our task is to achieve a reliable strategy to select the appropriate process noise covariance matrix, or specifically $Q_{rv,k}$ and $Q_{b,k}$, with the purpose of enhancing the filtering performance. Hence, the state space in the Q-learning approach is constructed based on the different design values of the process noise covariance matrix. Accordingly, every state of the two agents is related to an element in the per-determined sets $\left\{\begin{array}{ccc} \cdots & Q_{rv}^{\left(s_{rv}\right)} & \cdots \end{array}\right\}$ or $\left\{\begin{array}{ccc} \cdots & Q_{b}^{\left(s_{b}\right)} & \cdots \end{array}\right\}$, where $Q_{rv}^{\left(s_{rv}\right)}$ and $Q_{b}^{\left(s_{b}\right)}$ are the certain design values for $Q_{rv,k}$ and $Q_{b,k}$, respectively.

The action space is constructed to describe the different state transitions in the state space. To simplify the formulation of the algorithm and focus on the main task, in this paper, the action is set as remaining in the current state. To decrease the learning time cost, the deterministic Q-learning approach presented in [44] is adopted, where the Q-function for all of the states and actions is updated in each iteration. Alternative learning methods are described in [33] to explore the state space randomly with a certain action selection strategy, such as the ε-greedy strategy or Softmax strategy.

The utility functions of the two agents $U_{rv}\left(s_{rv},a_{rv}\right)$ and $U_{b}\left(s_{b},a_{b}\right)$ are designed based on the measurement innovation of the tentative EKFs where the related process noise covariance matrices $Q_{rv}^{\left(s_{rv}\right)}$ and $Q_{b}^{\left(s_{b}\right)}$ are adopted. As the measurement innovation is an effective indicator of the filtering performance, the values of $U_{rv}\left(s_{rv},a_{rv}\right)$ and $U_{b}\left(s_{b},a_{b}\right)$ are utilized to evaluate the quality of the related design values $Q_{rv}^{\left(s_{rv}\right)}$ and $Q_{b}^{\left(s_{b}\right)}$, and provide guidance to achieve the best possible strategy. The update laws of the two agents are formulated according to the iterative update equation of the Q-learning approach.

Following the previous description, for the navigation system model shown in (2) and (5), the detail implementation process of the PQEKF algorithm in one episode is presented in Algorithm 2.
**Algorithm 2**: Parallel Q-learning extended Kalman filter.**Input**$:\;\mathrm{initial}\;\mathrm{state}\;\mathrm{estimate}\;{\hat{x}}_{0}$$\;\mathrm{and}\;\mathrm{its}\;\mathrm{error}\;\mathrm{covariance}\;\mathrm{matrix}\;P_{0}$$,\;\mathrm{process}\;\mathrm{noise}\;\mathrm{covariance}\;\mathrm{matrix}\;Q_{k}$$,\;\mathrm{measurement}\;\mathrm{noise}\;\mathrm{covariance}\;\mathrm{matrix}\;R_{k}$$,\;\mathrm{measurement}\;\mathrm{sequence}\;\left\{y_{k}\right\}$ and initial Q-functions $Q_{rv}\left(s_{rv},a_{rv}\right)$ and $Q_{b}\left(s_{b},a_{b}\right)$
**Output**$:\;\left\{{\hat{x}}_{k}\right\}$$\;\mathrm{and}\;\left\{P_{k}\right\}$$1:\;\mathrm{for}\;\mathrm{all}\;s_{rv}\in S_{rv}$$,\;a_{rv}\in A_{rv}$, **do**$2:\;{\hat{x}}_{0}^{\left(s_{rv}\right)}\leftarrow {\hat{x}}_{0}$$,\;P_{0}^{\left(s_{rv}\right)}\leftarrow P_{0}$$,\;U_{rv}\left(s_{rv},a_{rv}\right)\leftarrow 0$$3:\;{\hat{Q}}_{k}^{\left(s_{rv}\right)}\leftarrow Q_{k}$$,\;{\hat{Q}}_{k}^{\left(s_{rv}\right)}(1:6,1:6)\leftarrow Q_{rv}^{\left(s_{rv}\right)}$4: **end**
$5:\;\mathrm{for}\;\mathrm{all}\;s_{b}\in S_{b}$$,\;a_{b}\in A_{b}$, **do**$6:\;{\hat{x}}_{0}^{\left(s_{b}\right)}\leftarrow {\hat{x}}_{0}$$,\;P_{0}^{\left(s_{b}\right)}\leftarrow P_{0}$$,\;U_{b}\left(s_{b},a_{b}\right)\leftarrow 0$$7:\;{\hat{Q}}_{k}^{\left(s_{b}\right)}\leftarrow Q_{k}$$,\;{\hat{Q}}_{k}^{\left(s_{b}\right)}(7:9,7:9)\leftarrow Q_{b}^{\left(s_{b}\right)}$8:  **end**9: for k=1,2,…,K$10:\;\mathrm{for}\;\mathrm{all}\;s_{rv}\in S_{rv}$$,\;a_{rv}\in A_{rv}$, **do**$11:\;[{\hat{x}}_{k}^{\left(s_{rv}\right)},P_{k}^{\left(s_{rv}\right)},{\tilde{y}}_{k}^{\left(s_{rv}\right)}]\leftarrow EKF({\hat{x}}_{k-1}^{\left(s_{rv}\right)},P_{k-1}^{\left(s_{rv}\right)},y_{k},{\hat{Q}}_{k}^{\left(s_{rv}\right)},R_{k})$$12:\;U_{rv}\left(s_{rv},a_{rv}\right)\leftarrow \left(1-K^{-1}\right)U_{rv}\left(s_{rv},a_{rv}\right)+K^{-1}{\left({\tilde{y}}_{k}^{\left(s_{rv}\right)}\right)}^{T}{\tilde{y}}_{k}^{\left(s_{rv}\right)}$$13:\;Q_{rv}\left(s_{rv},a_{rv}\right)\leftarrow U_{rv}\left(s_{rv},a_{rv}\right)+\gamma \underset{a_{\mathrm{rv}}^{\prime }\in A_{\mathrm{rv}}}{\min}Q_{rv}\left(s_{rv}^{\prime },a_{rv}^{\prime }\right)$14:   **end**$15:\;\mathrm{for}\;\mathrm{all}\;s_{rv}\in S_{rv}$$16:\;V_{rv}\left(s_{rv}\right)\leftarrow \underset{a_{\mathrm{rv}}\in A_{\mathrm{rv}}}{\min}Q_{rv}\left(s_{rv},a_{rv}\right)$17:   **end**$18:\;s_{rv,min}=arg\underset{s_{\mathrm{rv}}\in S_{\mathrm{rv}}}{\min}V_{rv}\left(s_{rv}\right)$$19:\;\mathrm{for}\;\mathrm{all}\;s_{b}\in S_{b}$$,\;a_{b}\in A_{b}$, **do**$20:\;[{\hat{x}}_{k}^{\left(s_{b}\right)},P_{k}^{\left(s_{b}\right)},{\tilde{y}}_{k}^{\left(s_{b}\right)}]\leftarrow EKF({\hat{x}}_{k-1}^{\left(s_{b}\right)},P_{k-1}^{\left(s_{b}\right)},y_{k},{\hat{Q}}_{k}^{\left(s_{b}\right)},R_{k})$$21:\;U_{b}\left(s_{b},a_{b}\right)\leftarrow \left(1-K^{-1}\right)U_{b}\left(s_{b},a_{b}\right)+K^{-1}{\left({\tilde{y}}_{k}^{\left(s_{b}\right)}\right)}^{T}{\tilde{y}}_{k}^{\left(s_{b}\right)}$$22:\;Q_{b}\left(s_{b},a_{b}\right)\leftarrow U_{b}\left(s_{b},a_{b}\right)+\gamma \underset{a_{b}^{\prime }\in A_{b}}{\min}Q_{rv}\left(s_{b}^{\prime },a_{b}^{\prime }\right)$23:    **end**$24:\;\mathrm{for}\;\mathrm{all}\;s_{b}\in S_{b}$$25:\;V_{b}\left(s_{b}\right)\leftarrow \underset{a_{b}\in A_{b}}{\min}Q_{b}\left(s_{b},a_{b}\right)$26:    **end**$27:\;s_{b,min}=arg\;\underset{s_{b}\in S_{b}}{\min}V_{b}\left(s_{b}\right)$$28:\;Q_{k}(1:6,1:6)\leftarrow Q_{rv}^{\left(s_{rv,min}\right)}$$,\;Q_{k}(7:9,7:9)\leftarrow Q_{b}^{\left(s_{b,min}\right)}$$29:\;[{\hat{x}}_{k},P_{k},{\tilde{y}}_{k}]\leftarrow EKF({\hat{x}}_{k-1},P_{k-1},y_{k},Q_{k},R_{k})$30: **end**


In the algorithm, ${\hat{x}}_{k}^{\left(s_{rv}\right)}$, $P_{k}^{\left(s_{rv}\right)}$, ${\hat{x}}_{k}^{\left(s_{b}\right)}$ and $P_{k}^{\left(s_{b}\right)}$ are the state estimates and the estimation error covariance matrices of the tentative EKFs for the two agents, ${\tilde{y}}_{k}^{\left(s_{rv}\right)}$ and ${\tilde{y}}_{k}^{\left(s_{b}\right)}$ are the corresponding measurement innovations, ${\hat{Q}}_{k}^{\left(s_{rv}\right)}$ and ${\hat{Q}}_{k}^{\left(s_{b}\right)}$ are the process noise covariance matrices for trial and K is the length of the measurement sequence in one episode. The Q-functions of the two parallel agents are updated with the utility functions $U_{rv}\left(s_{rv},a_{rv}\right)$ and $U_{b}\left(s_{b},a_{b}\right)$, which are accumulated in a certain time window to suppress the unfavorable effect of the measurement noise. Generally, if a relatively small utility function is obtained, the corresponding process noise covariance matrix for trial is considered to be satisfactory. Otherwise, the related ${\hat{Q}}_{k}^{\left(s_{rv}\right)}$ or ${\hat{Q}}_{k}^{\left(s_{b}\right)}$ are considered to be unsatisfactory. It is expected that the appropriate process noise covariance matrix, which is valuable to improve the filtering performance, is selected according to the Q-functions $Q_{rv}\left(s_{rv},a_{rv}\right)$ and $Q_{b}\left(s_{b},a_{b}\right)$ with this trial-and-error process. This algorithm can be implemented iteratively in multiple episodes to fine tune $Q_{k}$ in the navigation filter during the space mission.

For clarity, the structure of the presented PQEKF algorithm is shown in Figure 3.

For the implementation of the PQEKF, the measurement $y_{k}$ is acquired from the star sensors. Driven with the measurement data, the matrices $Q_{rv}^{\left(s_{rv,min}\right)}$ and $Q_{b}^{\left(s_{b,min}\right)}$ are selected with two parallel learning agents, which are designed based on the EKF algorithm and the Q-function update equation. With the appropriate process noise covariance matrix $Q_{k}$ composed of the sub-matrices $Q_{rv}^{\left(s_{rv,min}\right)}$ and $Q_{b}^{\left(s_{b,min}\right)}$, the navigation filter derives the optimized state estimate ${\hat{x}}_{k}$ for the spacecraft control system. Although the PQEKF presented here only contains two parallel learning agents, the algorithm can be extended easily for the case with multiple learning agents to deal with more complicated problems.

The presented PQEKF algorithm is suitable for the navigation systems where the prior knowledge about the statistical characteristics of the process noise or the measurement noise is insufficient. For example, in the relativistic navigation system, it is difficult to specify the magnitude of the process noise covariance related to the calibration parameter vector previously. To ensure the feasibility of the algorithm, before the implementation of the PQEKF on the orbit, all design values of the process noise covariance matrix could be tested through numerical simulations on the ground. For the considered system, a small state space in the Q-learning approach with a few design values of the process noise covariance on different orders of magnitude is sufficient to improve the filtering performance.

## 4. Simulations

### 4.1. Simulation Conditions

In this section, comparisons are performed to demonstrate the efficiency of the calibration method for the relativistic navigation system using the PQEKF. The reference trajectory of the spacecraft is generated through a high precision orbit propagator, where non-spherical Earth gravity perturbation, lunar–solar gravitational perturbation and solar radiation pressure perturbation are considered. Assume that the spacecraft is an MEO satellite in a near-circular orbit with a semi-major axis of 21,528 km and inclination of 55°. The measurement data are generated according to the reference trajectory and the measurement model shown in Section 2. The navigation filters designed based on the EKF and the PQEKF presented in Section 3 are implemented individually to process the measurement data. The position and velocity estimation errors are obtained via comparison between the state estimation and the reference trajectory.

For the fairness of comparison, the EKF and the PQEKF share the same measurement noise covariance matrix $R_{k}$ and the initial estimation error covariance matrix $P_{0}$. The parameter settings for the simulation are listed in Table 1.

For the PQEKF, when discretizing the state space, since the range of the state space is unknown, the upper limit and lower limit of the process noise covariance matrix is obtained through experiments. The performance of the presented methods is evaluated via the position and velocity estimation errors, which are critical for the orbital control of the spacecraft.

### 4.2. Simulation Results

First, the navigation performance of the presented method is compared with that of the traditional EKF without measurement bias calibration [24]. The three-axis position and velocity estimation error curves of the spacecraft obtained from the EKF without measurement bias calibration are shown in Figure 4 and Figure 5 with solid line, where the dashed lines represent the theoretic error bounds calculated from the estimation error covariance matrix of the navigation filter.

It is seen from Figure 4 and Figure 5 that the error curves fluctuate out of the theoretic error bounds frequently due to the unfavorable effect of the measurement bias. In contrast, the position and velocity estimation error curves of the calibration method based on the PQEKF are shown in Figure 6 and Figure 7.

From Figure 6 and Figure 7, it can be seen that all of the error curves are contained in the corresponding error bounds, which indicates the effectiveness for the design of the navigation filter.

Second, to facilitate the performance comparison of the algorithms in different simulation conditions, the position and velocity average root mean squared (RMS) errors of the EKF without bias calibration, the EKF with bias calibration and the presented method are plotted versus different settings of the measurement bias in Figure 8 and Figure 9.

It is easy to see from Figure 8 and Figure 9 that the estimation error of the EKF without bias calibration is enlarged with the increase in the measurement bias, while the effect of the measurement bias on the navigation performance is suppressed efficiently when the EKF with bias calibration is adopted. In addition, the calibration method based on the PQEKF achieves superior performance due to its ability in selecting the suitable process noise covariance matrix. It indicates that the presented method is not sensitive to the inter-star angle measurement bias.

In addition, the effect of the measurement noise on the PQEKF algorithm is examined through simulations. When the standard deviation of the measurement noise is changed in the scopes of [0.6, 1.6] mas, the RMS position errors of the EKF and the PQEKF are illustrated in Figure 10. The simulation result shows that, in comparison with the EKF, the PQEKF is more effective for suppressing the unfavorable effect of the measurement noise.

It is seen from Algorithm 2 that the PQEKF contains multiple EKFs. In the simulation, the execution time of the PQEKF is several times larger than that of the EKF. Nevertheless, it is easy to complete the one step iteration of the PQEKF algorithm in an observation period of the star sensors. For the considered system, to reduce the computational cost of the PQEKF, the state space or the action space of the Q-learning approach could be further optimized. For a complicated practical system with a large state space or action space, artificial neural network approximation or dedicated hardware can be introduced for the implementation of the algorithm. In addition, it is expected that a dynamic state space with the bound stretched automatically can be designed in future works.

Next, as the PQEKF is an improved version of the QLEKF, it is compared with the QLEKF for the relativistic navigation system through Monte Carlo trials. The position RMS error curves of the calibration methods based on the EKF, the AEKF, the QLEKF and the PQEKF are plotted in Figure 11. The statistical values of the navigation accuracy for the different methods are summarized in Table 2.

We can see from Figure 11 and Table 2 that the navigation performance of the PQEKF is slightly higher than the QLEKF, as two learning agents are implemented in parallel to select the appropriate $Q_{rv}^{\left(s_{rv,min}\right)}$ and $Q_{b}^{\left(s_{b,min}\right)}$ in parallel, while the QLEKF is designed to search for the whole $Q_{k}$. It is believed that the design of the PQEKF is more flexible than the QLEKF as different scale factors can be adopted to tune the different sub-matrices in the process noise covariance matrix.

Finally, the influence of the state space discretization on the positioning accuracy of the PQEKF algorithm for the relativistic navigation system is analyzed. When the number of states is set as five, seven and nine, respectively, the position RMS error curves of the PQEKF algorithms are shown in Figure 12.

From Figure 12, the variation in the position estimation error under the different settings of the state number in the state space discretization is rather small in the majority of the simulation processes. This indicates that the influence of the state number variation within a certain scope on the estimation accuracy of the PQEKF is not evident. In the considered scenario, two pre-determined sets with a small number of design values for $Q_{rv}$ and $Q_{b}$ are beneficial for improving the performance of the calibration method.

According to the above simulation analysis, it is confirmed that the presented method is well-suited for the relativistic navigation system with the requirement to calibrate the inter-star angle measurement bias. For the simulation conditions described in Section 4.1, the achievable spacecraft navigation accuracy is on the order of a few hundred meters, which is sufficient for most orbital control missions.

## 5. Conclusions

This paper presents an inter-star angle measurement bias calibration method for the spacecraft relativistic navigation system. The proper design of the process noise covariance matrix is critical for accurate calibration. In order to improve the calibration accuracy and the navigation performance, the Q-learning approach is combined with the EKF for an online adaptive tuning of the process noise covariance matrix based on the measurement data achieved from onboard star sensors. The PQEKF algorithm is developed as the navigation filter, where two learning agents are implemented in parallel to select the appropriate sub-matrices related to the kinematic state and the calibration parameters, respectively. The simulation results show that the navigation performance of the presented method is superior to that of the EKF, the AEKF and the QLEKF in the presence of measurement bias, demonstrating the efficiency of the calibration method and the PQEKF algorithm. This study introduces a hybrid framework to combine the reinforcement learning approach in the navigation filter, which can serve as a foundation method to improve the state estimation accuracy in potential applications of relativistic navigation for Earth satellites or deep space explorers.

## Figures and Tables

**Figure 1 sensors-24-06186-f001:**
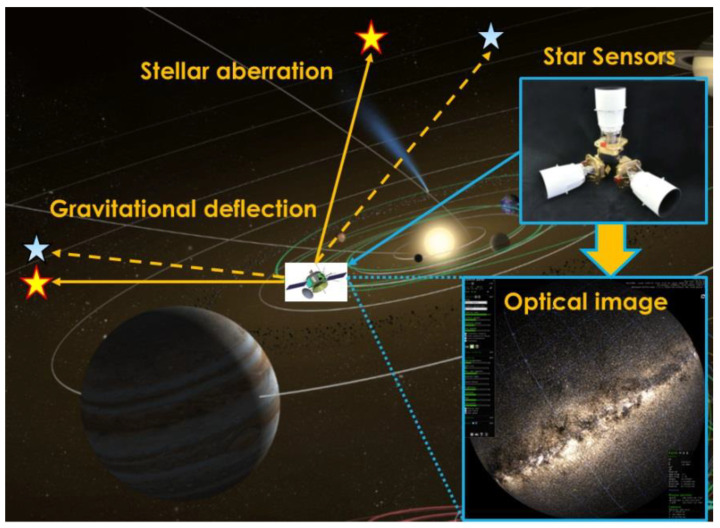
Concept of relativistic navigation.

**Figure 2 sensors-24-06186-f002:**
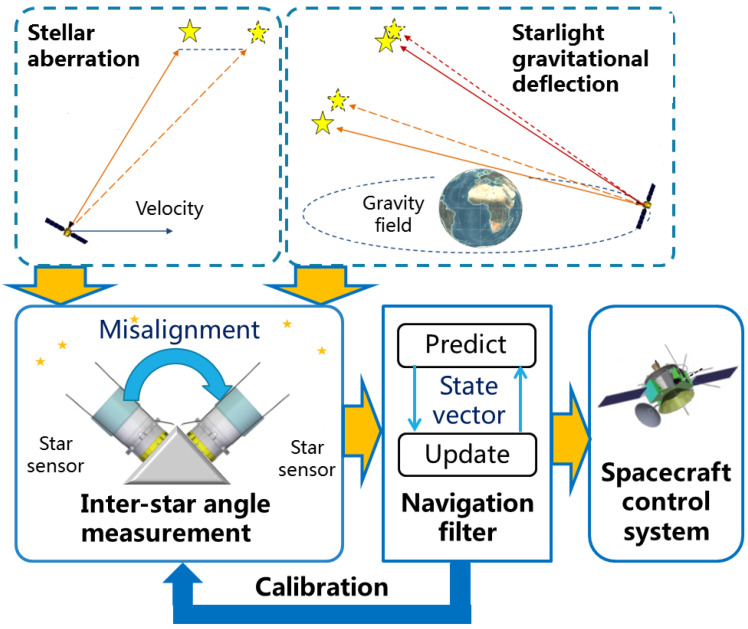
Diagram of relativistic navigation and calibration method.

**Figure 3 sensors-24-06186-f003:**
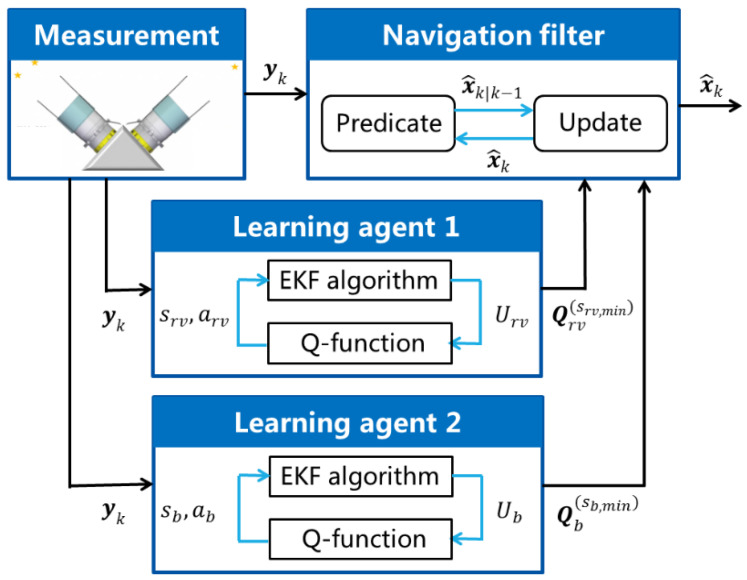
Diagram of PQEKF algorithm.

**Figure 4 sensors-24-06186-f004:**
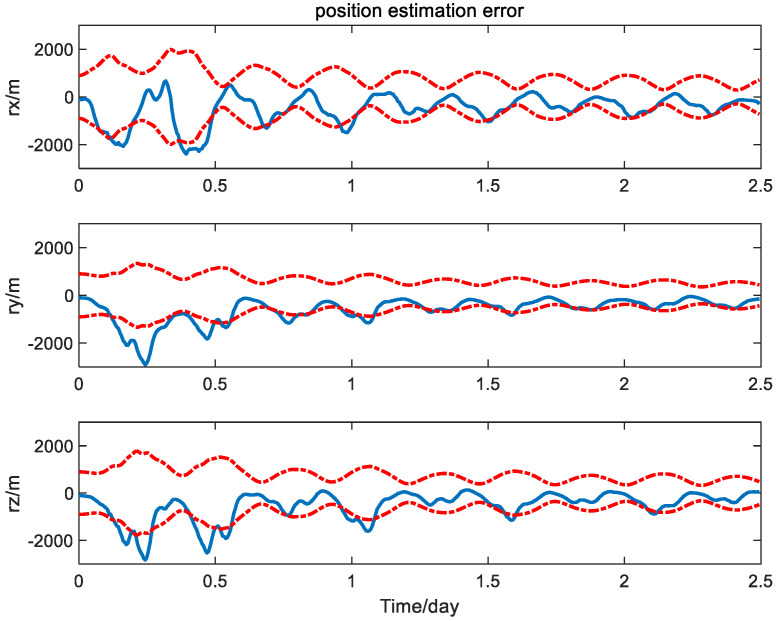
Position estimation error of traditional EKF without measurement bias calibration.

**Figure 5 sensors-24-06186-f005:**
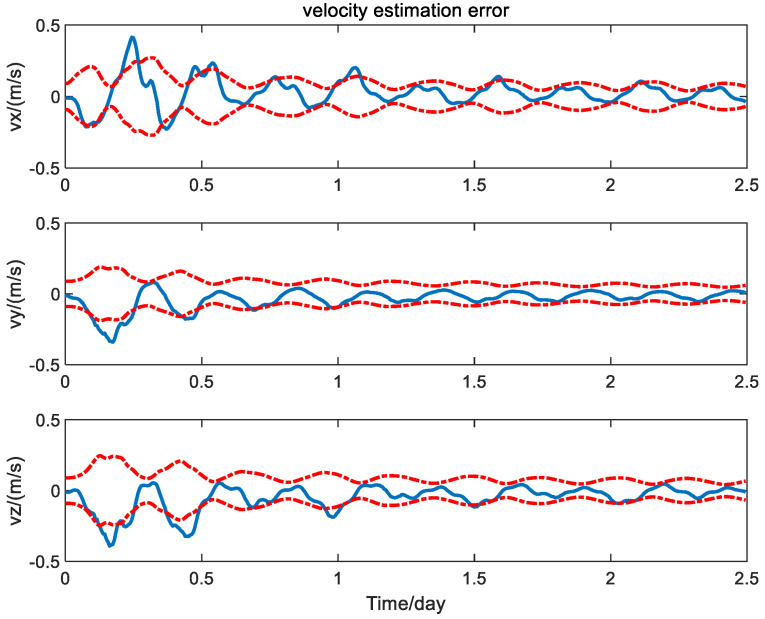
Velocity estimation error of traditional EKF without measurement bias calibration.

**Figure 6 sensors-24-06186-f006:**
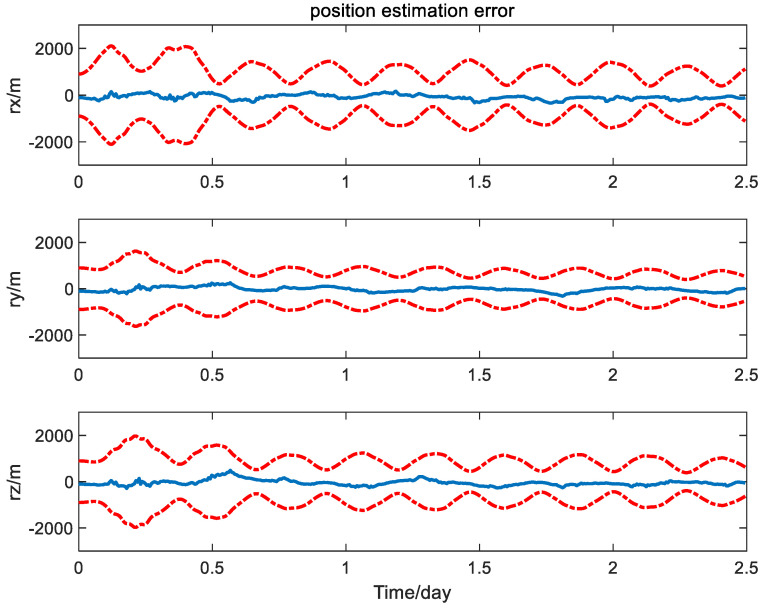
Position estimation error of calibration method based on PQEKF.

**Figure 7 sensors-24-06186-f007:**
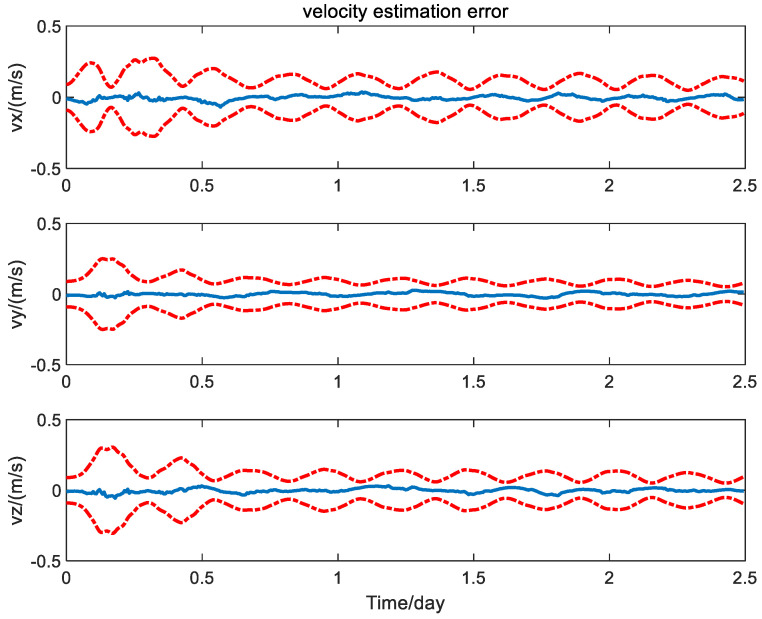
Velocity estimation error of calibration method based on PQEKF.

**Figure 8 sensors-24-06186-f008:**
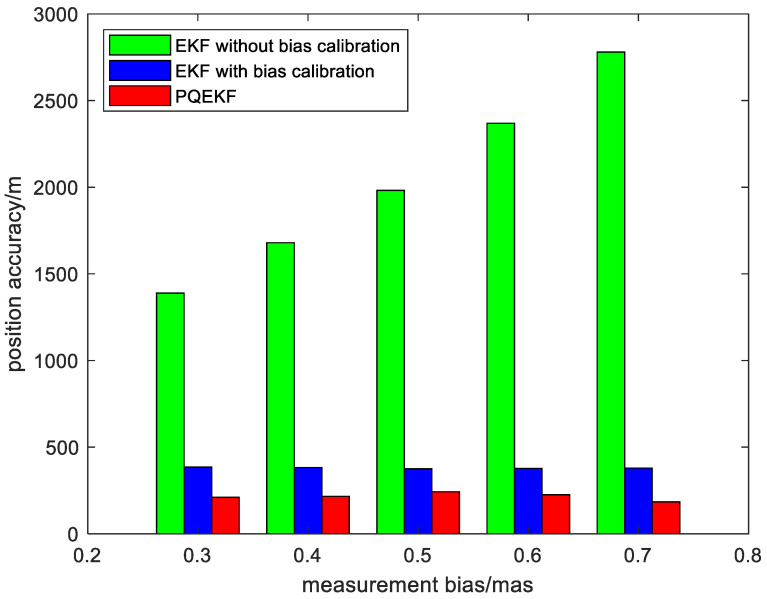
Position RMS errors of different methods vs. measurement bias.

**Figure 9 sensors-24-06186-f009:**
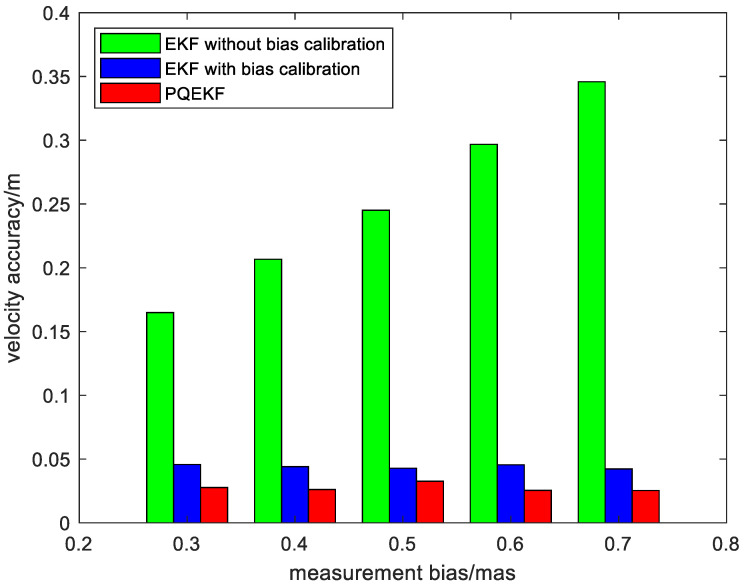
Velocity RMS errors of different methods vs. measurement bias.

**Figure 10 sensors-24-06186-f010:**
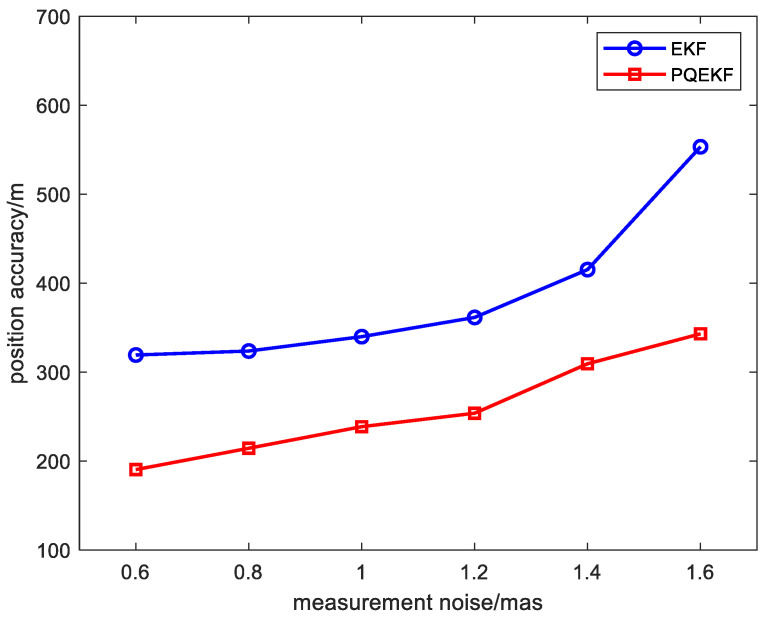
RMS errors as functions of measurement noise standard deviation.

**Figure 11 sensors-24-06186-f011:**
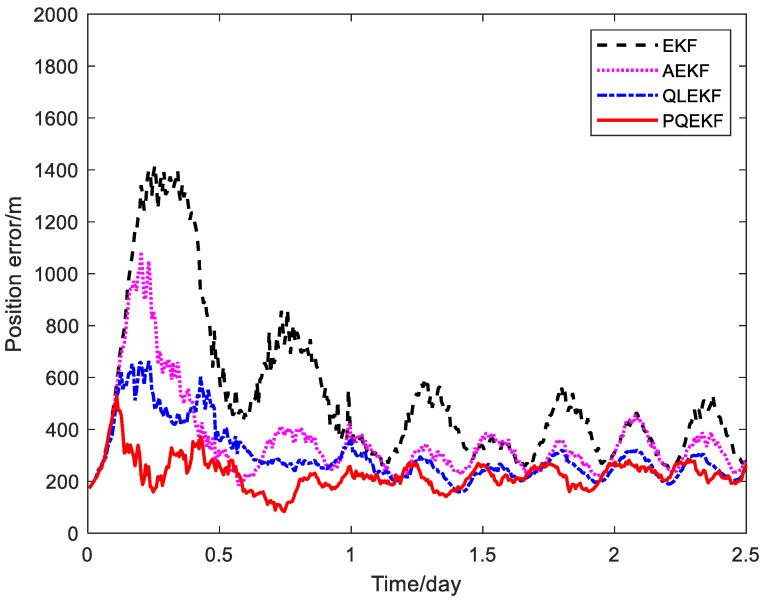
Position RMS error curves of different navigation filters.

**Figure 12 sensors-24-06186-f012:**
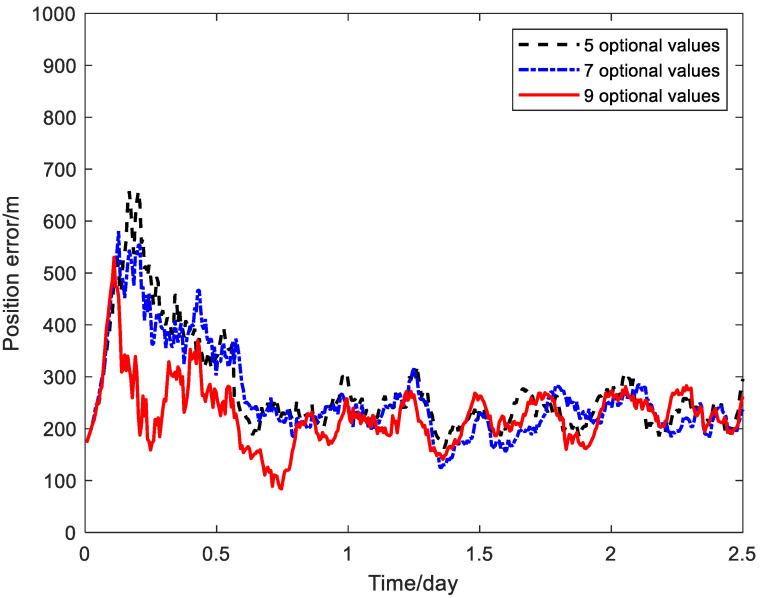
Position RMS error curves of PQEKF algorithms for different state numbers.

**Table 1 sensors-24-06186-t001:** Simulation parameter settings.

Simulation conditions	Duration of simulation	2.5 days
	Measurement noise standard deviation	1 mas
	Measurement bias	0.3 mas
	Update frequency	0.1 Hz
EKF parameters	Initial estimation error covariance	$P_{0}=diag([p_{r},p_{r},p_{r},p_{v},p_{v},p_{v},p_{b},p_{b},p_{b}])$ $p_{r}=300\;m,p_{v}=0.03\;m/s,p_{b}=0.1\;mas$
	Process noise covariance	$Q_{k}=diag([q_{r},q_{r},q_{r},q_{v},q_{v},q_{v},q_{b},q_{b},q_{b}])$ $q_{r}=1\times 10^{-5}\;m,q_{v}=1\times 10^{-5}\;m/s,$ $q_{b}=0.03\;mas$
	Measurement noise covariance	$R_{k}=diag([\sigma_{ISA},\sigma_{ISA},\sigma_{ISA}])$ $\sigma_{ISA}=1\;mas$
PQEKF parameters	State space for agent 1	$\left\{15^{-2}Q_{rv,k},10^{-2}Q_{rv,k},5^{-2}Q_{rv,k},Q_{rv,k},5^{2}Q_{rv,k},10^{2}Q_{rv,k},15^{2}Q_{rv,k}\right\}$ $Q_{rv,k}=Q_{k}(1:6,1:6)$
	State space for agent 2	$\left\{150^{-2}Q_{b,k},100^{-2}Q_{b,k},50^{-2}Q_{b,k},Q_{b,k},50^{2}Q_{b,k},100^{2}Q_{b,k},150^{2}Q_{b,k}\right\}$ $Q_{b,k}=Q_{k}(7:9,7:9)$
	Window size	K=50
	Discounted factor	γ=0.9

**Table 2 sensors-24-06186-t002:** Comparison of calibration methods based on EKF, AEKF, QLEKF and PQEKF.

Calibration Method	Average RMS Error		
	Position (m)	Velocity (m/s)	Measurement Bias (mas)
EKF	609.8	0.073	0.275
AEKF	371.1	0.045	0.187
QLEKF	328.3	0.036	0.088
PQEKF	215.5	0.025	0.055

## Data Availability

Data are contained within the article.

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
