# Peer review of "Calibration Method for Relativistic Navigation System Using Parallel Q-Learning Extended Kalman Filter"

_sensors, 2024, doi:10.3390/s24196186_

Round 1

Reviewer 1 Report

Comments and Suggestions for Authors

The manuscript aims to develop a calibration method for relativistic navigation systems using an extended Kalman filter with reinforcement
learning (Q-learning). Its focus is on achieving better precision in the
autonomous navigation of spaceships, where relativistic perturbations
such as stellar aberration and gravitational deflection of light affect angle measurements between stars.

The proposed method, Parallel Q-Learning Extended Kalman Filter (PQEKF), is designed to calibrate the measurement bias caused by the misalignment of stellar sensors, which is one of the main sources of error in relativistic navigation systems. The core innovation is the use of two parallel learning agents for the adaptive adjustment of the noise correlation matrix of the process, thus enhancing the precision of the estimated position and velocity of the spaceship. Numerical simulations performed in a Medium Earth Orbit (MEO) satellite scenario show that this method achieves a significant reduction in navigation errors, with precision reaching a few hundred meters.

The manuscript concludes that the PQEKF method is effective in improving the performance of autonomous navigation systems and could be used in future space missions. The text is clear and well-structured and presents innovations, but it could be improved by including a deeper discussion on the technical choices, practical challenges of implementation, and the scalability of the proposed methodology.

Moreover, including simulations in different scenarios and discussing
experimental validation would add robustness to the proposal and help
pave the way for real-life applications. For instance, in Section 3.3, the authors might discuss the additional computational complexity introduced by the parallel execution of two Q-learning agents, possibly by analyzing execution times or computational costs. They might also explain how this approach could be adapted for embedded systems with less computational power.

Finally, the authors might consider mentioning possible optimizations of the approach to reduce the computational effort required, such as its implementation in dedicated hardware (FPGA, ASIC) or approximation methods to minimize the volume of calculations. I believe that, after addressing these considerations, the manuscript could be suitable for publication.

Reviewer 2 Report

Comments and Suggestions for Authors

Please check Equation no. 32 in line no. 321

Comments on the Quality of English Language

Last line of Abstract can be rewritten.

Reviewer 3 Report

Comments and Suggestions for Authors

This paper sounds intriguing. Computational and mathematical methods are used for analysis and calibration using a Q-Learning Extended Kalman Filter. In my opinion, the study has useful material; however, it needs to be restructured for better understanding. When the Q-learning approach is aligned with EKF from onboard star sensors and PQEKF algorithm is used as the navigation filter, the transition between the two is unclear in terms of which capacity interacts with the other. The reader will benefit from a clarification of how both are integrated.

It looks like there is not enough background/data on EKF parameters when designing an algorithm. What exactly is the covariance matrix used for?

It is relevant but it needs more literature on Q-learning filters.

Develop a smoother correlation between the process noise covariance matrix and navigation filter algorithm to improve navigation performance. Consider using an artificial neural network to explain inbound and outbound actions.

Reviewer 4 Report

Comments and Suggestions for Authors

This paper proposes a novel parallel Q-learning extended Kalman filter (PQEKF) for calibrating measurement biases in relativistic navigation systems, which is a very interesting and challenging area of research.

After reading the full text carefully, I think this paper has made a valuable contribution. However, in order to improve the quality and readability of the paper, I recommend that the author consider the following comments:

1. The PQEKF algorithm proposed is one of the core contributions. It is suggested that the authors can provide explanations about the principle of Q learning in the covariance matrix of the adjustment process noise. For the convergence and stability of the algorithm, it is recommended to provide more rigorous mathematical proofs or theoretical analysis. If it is proposed in appendix, please remark it in context.

2. The experimental part is the key to validating the performance of the algorithm. Authors are advised to provide more details, including the experimental setup, data sources, and parameter settings used in the experiment. Especially, the sources and channels of the dataset in the simulations should be announced in the work. It is suggested the comparisons with traditional methods should be in one identical figure.

3. The authors are advised to expand the literature review section, especially context related with application of relativistic navigation system, and algorithms in the relativistic navigation method. This will help readers better understand the background and importance of the study.

4. For algorithm flow and system models, it is recommended to use flowcharts or block diagrams to visualize them to help readers better understand how algorithms work.

5. The writing quality of the essay is generally good, but some parts, like Introduction, and conclusion, may need further polish to improve the fluency and accuracy of the language.

6. It is suggested that the authors discuss future research directions in the conclusion section, including possible improvements or potential applications.

7. The algorithms in contexts and appendix section can be simplified. The authors have to ensure that these are not necessary to understand the main results or are appropriately cited in the body.

After the authors revise the above recommendations, I believe that this paper will contribute to the field of relativistic navigation system.
